# Chromosome-Level Genome Assembly of *Papilio elwesi* Leech, 1889 (Lepidoptera: Papilionidae)

**DOI:** 10.3390/insects14030304

**Published:** 2023-03-21

**Authors:** Zhixiang Pan, Yinhuan Ding, Shusheng Zhang, Luxian Li, Fangzhou Ma

**Affiliations:** 1School of Life Sciences, Taizhou University, Taizhou 318000, China; 2Department of Agronomy and Horticulture, Jiangsu Vocational College of Agriculture and Forestry, Jurong 212400, China; 3The Management Center of Wuyanling National Natural Reserve in Zhejiang, Wenzhou 325500, China; 4Zhejiang Environment Technology Company Limited, Hangzhou 311100, China; 5Nanjing Institute of Environmental Sciences under Ministry of Ecology and Environment, Nanjing 210042, China

**Keywords:** butterfly, Papilioninae, gene family evolution, genome synteny, comparative genomics

## Abstract

**Simple Summary:**

Swallowtail butterflies are renowned for their extensive morphological diversity, especially their wing color patterns, with around 600 described species. Given their morphological diversity and species richness, as well as their ecological habits, their phytophagous insect lineage is regarded to have evolved successfully. According to the NCBI database, sixty Papilionidae genomes have been reported, of which only two species have been assembled at the chromosome level. Currently, the paucity of high-quality genome data, especially for rarely seen and endemic butterflies, has hampered our knowledge of their evolution and biology at the genome level. Here, we report a chromosome-level genome assembly of *Papilio elwesi*, a butterfly species endemic to the Chinese mainland. The sequencing, assembly, and annotation of *Papilio elwesi* in this study enrich the available butterfly genome resources and lay the data foundation for future research.

**Abstract:**

A rarely seen butterfly species, the large swallowtail butterfly *Papilio elwesi* Leech, 1889 (Lepidoptera: Papilionidae), endemic to the Chinese mainland, has been declared a state-protected animal in China since 2000, but its genome is not yet available. To obtain high-quality genome assembly and annotation, we sequenced the genome and transcriptome of *P. elwesi* using the PacBio and PromethION platforms, respectively. The final assembled genome was 358.51 Mb, of which 97.59% was anchored to chromosomes (30 autosomes and 1 Z sex chromosome), with a contig/scaffold N50 length of 6.79/12.32 Mb and 99.0% (*n* = 1367) BUSCO completeness. The genome annotation pointed to 36.82% (131.99 Mb) repetitive elements and 1296 non-coding RNAs in the genome, along with 13,681 protein-coding genes that cover 98.6% (1348) of the BUSCO genes. Among the 11,499 identified gene families, 104 underwent significantly rapid expansions or contractions, and these rapidly expanding families play roles in detoxification and metabolism. Additionally, strong synteny exists between the chromosomes of *P. elwesi* and *P. machaon*. The chromosome-level genome of *P. elwesi* could serve as an important genomic resource for furthering our understanding of butterfly evolution and for more in-depth genomic analyses.

## 1. Introduction

Among the 18,000+ known butterfly species (Lepidoptera: Papilionoidea), the family Papilionidae Latreille 1802, commonly known as swallowtail butterflies, comprises more than 3% of all butterfly species [1]. Owing to the colorful wing color patterns and the extensive morphological diversity among the species, sexes, populations, and seasonal forms [2,3,4], it is widely regarded as one of the most popular and esthetically attractive butterflies and of great significance for exploring the morphological diversification and speciation of butterflies [5,6]. According to Espeland et al. [7], Papilionidae was phylogenetically placed at the basal position as a sister group to all remaining butterflies. Moreover, swallowtail butterflies are one of the most widely studied groups of insects and are considered model organisms in a number of fields, including genetics, ecology, evolutionary biology, conservation biology, development, etc. [8,9,10,11,12]. As a result of developments in sequencing technology, samples can now be directly collected from the wild for sequencing and subsequent analysis, bypassing some of the time-consuming processes in laboratory rearing and inbreeding [13,14,15,16,17]. This provides great possibilities to learn more about butterflies, particularly for some uncommon and rare species.

The explosion of available genomic data has enabled our research to exploit single genes to entire genomes. As genomic data are increasingly being used to address questions from a wide range of research areas, the completeness (e.g., BUSCO scores) and continuity (e.g., N50 length) of genomes have also become vitally important [18]. Ellis et al. [19] have investigated all available genomic data (assemblies and raw sequencing data) of butterflies that were deposited in the LepBase [20] and NCBI [21] databases before July 2020 and have evaluated their quality according to the completeness and continuity of the draft genome assemblies, showing that most assemblies were at the scaffold or contigs level, and a portion of them was unusable due to fragmentation (<200 bp) or contamination, with only three papilionoid species assembled at the chromosomal level. To date, there are only 53 chromosome-level butterfly genomes in the NCBI (as of 6 March 2023), less than 0.3% of the total number of butterflies. Among them, 29 are of Nymphalidae, 11 are of Lycaenidae, 11 are of Pieridae, and 2 are of Papilionidae. More chromosome-level genomes of butterflies may provide an opportunity to identify elements that regulate genetic variation in morphological traits [22,23,24], and may make it easier to conduct research on the evolutionary biology of butterflies, such as sex chromosome evolution [25,26,27].

*Papilio elwesi* Leech, 1889 (Papilionidae: Papilioninae), a large broad-tailed swallowtail butterfly, is endemic to the Chinese mainland. It is distributed from southwest to southeast China and inhabits sparsely populated deep mountains and dense forests. The distribution of *P. elwesi* is related to its hosts, as its larvae have a relatively homogeneous diet, feeding only on some plants of the Magnoliaceae and *Sassafras tzumu* of the Lauraceae. Due to its specific habitat and feeding habits, it is very rare to see this butterfly in general, let alone obtain specimens of it. As a state-protected animal in China since 2000, *P. elwesi*, has received a great deal of public attention, but it is debatable as to its phylogenetic position. There are very few publicly available genetic data on it, with only a few sequenced barcodes. Since there is no public genome assembly information about *P. elwesi*, this substantially restricts the scope of its future applicability. Although there are 30 genome assemblies of 21 *Papilio* species available in the NCBI database (accessed 6 March 2023), only a single chromosome-level genome of *Papilio machaon* has been public. Having a high-quality reference genome will offer a reliable genetic foundation for comprehending the molecular mechanisms, physiological processes, and evolutionary biology associated with adaptations to environmental change [28]. Comparing the genome of the rarely seen Chinese broad-tailed swallowtail butterfly *P. elwesi* to other lepidopteran genomes can help in understanding the genetic, evolutionary, and developmental mechanisms of butterflies.

In this study, we applied multiple platforms to sequence HiFi, ONT, and Hi-C data of the butterfly *P. elwesi,* primarily to exploit their respective strengths and to improve the quality of subsequent genome assembly and annotation. The genome of *P. elwesi* was assembled at the chromosome level and annotated with high quality. We also assembled and annotated its mitochondrial genome. Gene family evolution and phylogenetic reconstruction analyses were further performed, as was interspecific chromosomal variation analysis.

## 2. Materials and Methods

### 2.1. Sample Collection and Sequencing

Two live pupae of *P. elwesi* were collected on 2 October 2021, from *Sassafras tzumu* in Xianju National Park, Xianju County, Taizhou City, Zhejiang Province, China. We chose pupae rather than adult specimens to acquire purer DNA and RNA while minimizing the risk of potential digestive and other microbial contamination. The pupae were delivered to Berry Genomics (Beijing, China), and then stored in liquid nitrogen for subsequent DNA and RNA extraction, library preparation, and sequencing. One pupal individual was used for PacBio HiFi and Illumina transcriptome sequencing, whereas Illumina whole-genome (genome survey), Hi-C (genome), and ONT (transcriptome) sequencing were performed on the other pupa. Genomic DNA and transcriptome RNA were extracted by the cetyltrimethylammonium bromide (CATB) method and TRIzol™ Reagent, respectively. Prior to sequencing, quality control was performed on the extracted DNA and RNA to ensure that they met the requirements of the sequencing platforms. The quality of the DNA was checked using the NanoDrop 2000 Spectrophotometer, Qubit fluorometer, pulsed-field electrophoresis (Bio-rad CHEF), and agarose gel electrophoresis, while the RNA was monitored using the Agilent 4200 TapeStation system and NanoDrop 2000 Spectrophotometer. All details about the sequencing platform, insert size, and data amount are summarized in Appendix A.

### 2.2. Genome Survey and Assembly

Quality control was performed using Fastp v0.23.0 [29] with parameters “–q 20 –D –g –x –u 10-5 –r –c” to eliminate bases with quality scores below 20, drop duplication, trim polyG/X tails, and enable base correction in overlapped regions. The script khist.sh in BBTools v38.82 [30] was used to generate a histogram file of 21-kmers for the genome survey in GenomeScope2 v2.0.0 [31], and the genome size estimation for *P. elwesi* was set to “–k 21 –p 2 –m 10000”.

Prior to the genome assembly, a format conversion of PacBio HiFi ccs reads from BAM to FASTA was performed in SAMtools v1.10 [32]. Hifiasm v0.16.1-r375 [33] was used for the preliminary genome assembly, and Gfatools converted the assembly graph of the primary contigs in the GFA format to the FASTA file. Contigs of read coverage lower than 20× were filtered. Minimap2 v2.24-r1122 [34,35] was used for mapping reads and Purge_Dups v1.2.5 [36] was used to delete redundant heterozygous regions.

Scaffolding with Hi-C data employed Juicer v1.6.2 [37] and 3D-DNA v180922 [38]; the former was used to align reads to the assembly, remove duplicates, and extract Hi-C contacts, the latter was used for anchoring primary contigs into chromosomes. Then we manually reviewed the candidate assembly in conjunction with Juicebox v1.11.08 [37] according to the Hi-C heatmaps. Additionally, compared to the UniVec and nucleotide databases in the NCBI, a blastn-like search was performed using MMseqs2 v11 [39] for detecting potential contaminant sequences within the assembly. Further assessments of the genome quality were carried out based on the raw reads mapping rate, consensus quality (QV) estimation, and genome completeness, calculated using SAMtools, Merqury v1.3 [40], and BUSCO v5.2.2 [41], respectively.

In addition, the mitochondrial genome of *P. elwesi* was assembled by NOVOPlasty v4.3.1 [42] employing Illumina genome short reads, with the cytochrome oxidase subunit I (COI) gene of *P. elwesi* deposited in Genbank (JQ086720.1) as the seed, and followed by MitoZ_v2.4-alpha [43] for annotation.

### 2.3. Genome Annotation

Genomes are usually annotated with repetitive sequences, protein-coding genes (PCGs), and non-coding RNAs (ncRNAs). RepeatModeler v2.0.3 [44], used to mask repeats in the assembled genome, enabled the LTR discovery pipeline (-LTRStruct) to create a de novo repeat library specific for *P. elwesi*. The library was then merged with the repetitive DNA elements database RepBase-20181026 [45] into a custom library for identifying repetitive elements by RepeatMasker v.4.1.2 [46]. Infernal v1.1.4 [47] was used to identify ncRNAs against the RNA families database Rfam v14.8 [48], and tRNAs were further predicted and confirmed with tRNAscan-SE v.2.0.9 [49].

Protein-coding gene prediction integrated evidence at the DNA, RNA, and protein levels through the MAKER v.3.01.03 [50] pipeline, which can be boiled down into three strategies. The first strategy was ab initio gene prediction using protein and transcriptome evidence in BRAKER v2.1.6 [51], where Augustus v3.4.0 [52] was combined with GeneMark-ES/ET/EP v4.68_lic [53] as predictors integrated for gene model training; the reference Arthropoda protein evidence was downloaded from the OrthoDB10 protein database [54] and incorporated into BRAKER along with RNA-seq data to increase the coding gene prediction accuracy. The second strategy was transcriptome-based prediction, where transcripts (transcriptome evidence) were assembled using StringTie v.2.1.3 [55] with mixed Illumina short reads and ONT reads. HISAT2 v.2.2.0 [56] was used to align RNA short reads to the assembled *P. elwesi* genome, while Minimap2 was used to map ONT reads. The third strategy was homology-based prediction, for which we downloaded high-quality genome assemblies and annotations of the following six reference species: *Drosophila melanogaster* (Diptera), *Spodoptera frugiperda* (Noctuoidea), *Aricia agestis* (Lycaenidae), *Danaus plexippus* (Nymphalidae), *Pieris rapae* (Pieridae) and *Papilio machaon* (Papilioninae) as evidence of protein homology, and combined them with transcriptome evidence for gene predictions in GeMoMa v1.8 [57]. Using the evidence presented above, MAKER produced the final genome structure annotation.

Annotations describe features of the genome, not only structurally but also functionally. Gene function annotation was conducted using three tools, including eggNOG-mapper v2.1.5 [58], Diamond v2.0.11.149 [59], and InterProScan v5.53-87.0 [60]. The eggNOG-mapper searched the Diamond-compatible eggNOG v5.0 [61] database in the very-sensitive mode to identify the KEGG pathway, gene ontology (GO), etc. Homology-based functional genes were assigned by Diamond searching against the UniProtKB (SwissProt+TrEMBL) databases. InterProScan searched multiple databases (Pfam [62], Smart [63], Superfamily, [64], and CDD [65]) to output protein domains, GOs, Reactome pathways, etc. The results predicted by the above tools were integrated to obtain the final gene function prediction.

### 2.4. Gene Family Identification and Evolution

Using OrthoFinder v2.5.2 [66], orthology was inferred across the *Bombyx mori* (Bombycidae), *Spodoptera frugiperda* (Noctuidae), *Pieris rapae* (Pieridae), *Danaus plexippus* (Nymphalidae), *Aricia agestis* (Lycaenidae), and six *Papilio species* (*P. bianor*, *P. elwesi*, *P. glaucus*, *P. machaon*, *P. polyte*, and *P. xuthus*). The protein sequences of 10 species were all downloaded from the NCBI with redundant isoforms removed, and then fed into OrthoFinder, employing Diamond for sequence alignment.

To infer the phylogeny, we extracted single-copy ortholog amino acid sequences and applied custom scripts of phylogenomics from GitHub (https://github.com/xtmtd/Phylogenomics/tree/main/scripts accessed on 23 August 2022) to align (align_MAFFT.sh), trim (trimming_alignments.sh), filter (loci_filtering_alignment-based.sh), and concatenate (matrix_generation.sh) them. MAFFT v7.487 [67] was used to execute multiple sequence alignment (MSA) using the option “L-INS-i” to increase accuracy. BMGE v1.12 [68] was used for trimming MSAs with a large BLOSUM matrix value of 90 to increase stringency. We used PhyKIT v1.11.10 [69] to filter loci based on the number of parsimony-informative sites and composition heterogeneity (RCV) and then concatenated them into a matrix. Following that, the best-fit partition scheme and model were searched by ModelFinder [70], which specified nuclear of general amino acid models (–msub nuclear) and was restricted to a subset of LG models (–mset LG). Only the partitioning schemes at the top 10% (–rclusterf 10) were selected for phylogeny inference in IQ-TREE v2.1.3 [71], with 1000 replicates of the ultrafast bootstrap (–B 1000) and SH-aLRT test (–alrt 1000).

Divergence time estimation was performed by using the script mcmctree_AA.sh (https://github.com/xtmtd/Phylogenomics/tree/main/scripts accessed on 23 August 2022), which applies MCMCTree from PAML v.4.9j [72]. Four fossil calibration points from Espeland et al. [7] were selected here: root (<201.3 Ma), Papilionoidea (91.0–143.0 Ma), Nymphalidae (71.0–112.0 Ma), and Lycaenidae (60.0–96.0 Ma). To guarantee convergence, the script was run at least twice. At last, changes (expansion or contraction) in the genome families were inferred across the divergence time estimation tree using CAFE v.4.2.1 [73], which applied an error correction model and lambda (birth–death parameter) search using the default parameters. Functional enrichment analysis based on PCGs from significantly expanded families enriched the GO and KEGG terms using clusterProfiler v3.10.1 [74], and the cutoff values of the p-value and q-value were respectively set to 0.01 and 0.05, with the Benjamini–Hochberg (BH) correction method.

### 2.5. Chromosomal Synteny

To reveal the interspecific chromosomal variation between *P. elwesi* and *P. machaon*, MMseqs2 was searched against their protein sequence alignments using the easy-search module with a target sensitivity of 7.5, a maximum E-value threshold of 1e-5, and 5 hits accepted per query sequence (–s 7.5 –alignment-mode 3 –num-iterations 1-e 1e-5 –max-accept 5). Then, we obtained the synteny between two papilionid species using MCXcanX [75] with an alignment significance level of 1e-10 (–e 1e-10) and a syntenic block of five genes (–s 5).

## 3. Results and Discussion

### 3.1. Sequencing and Genome Survey

There was a total of 147.24 Gb data sequenced for *P. elwesi* (Appendix A), including 123.05 Gb (32.13 Gb PacBio HiFi, 54.51 Gb Illumina, 36.41 Gb Hi-C) of DNA and 24.19 Gb (11.58 Gb Illumina, 12.61 Gb ONT) of RNA. The mean/N50 lengths of the HiFi and ONT reads were 16.61/16.51 kb and 1.11/1.45 kb (Appendix A), respectively. For the 54.51 Gb Illumina DNA data, 40.06 Gb was retained after the quality control process and then used for the genome survey. The genome was estimated to be 331.80–331.96 Mb in size, with 0.93% heterozygosity and 17.78% (58.99–59.02 Mb) repetitive sequences. The peak (~44×) in front of the main peak (~88×) may be the sex chromosome, which accounts for about half of the coverage of the main peak (Appendix A).

### 3.2. Genome and Mitochondrion Assembly

The final *P. elwesi* assembly contained 147 scaffolds and 207 contigs, the N50 length of the scaffold/contig was 12.32/6.79 Mb, and the length of the max scaffold/contig was 20.18/15.81 Mb (Appendix A). Its size was 358.51 Mb, with 97.59% anchored on 31 chromosomes (Figure 1a and Appendix A), and its GC content was 37.08% (Appendix A). The sequencing coverage of each chromosome was calculated and is shown in Appendix A. The coverage of one chromosome is 45.86×, which is approximately half the coverage of most chromosomes. Since a few Lepidoptera females have no W sex chromosome, the sex chromosome system of Lepidoptera is WZ/Z for females and ZZ for males [26], and both the Z and W sex chromosomes for females are theoretically half the coverage of the autosomes. However, only one chromosome was sequenced about 45× here. We judged it to be sex chromosome Z, and the *P. elwesi* we collected was a female swallowtail butterfly. The genome survey estimated a smaller genome size because only the autosomal size was calculated. The coverage of some chromosomes is apparently higher than 90×, which is because of the high proportion of transposons (repetitive sequences), resulting in multiple reads repeatedly mapped to one region. The BUSCO assessment indicated that the completeness of the assembled genome is up to 99.0% (insecta_odb10, *n* = 1367) with 98.2% single-copy orthologues and 0.8% duplicated genes. The raw sequencing data mapped to the assembly had a high mapping ratio: 99.9%/95.07% for long/short DNA and 91.96% for short RNA, and the estimated consensus quality value for the assembly reached 55 (QV = 55). These results indicate that the genome we assembled is of high quality.

In comparison to the two Papilionidae chromosome-level assembled genomes available in the NCBI database, the genome size of *P. elwesi* is bigger than that of *P. machaon* (GCA_912999745.1, 252.11 Mb) but smaller than that of *Iphiclides podalirius* (GCA_933534255.1, 430.73 Mb). It has a much longer scaffold N50 length than the prior assembly of *P. machaon* (8.78 Mb) and the highest GC content. In terms of the BUSCO completeness, it is on par with *P. machaon* at 99% and higher than *Iphiclides podalirius* at 97% (Appendix A). It is evident from the high degree of contig contiguity and completeness that the assembled genome of *P. elwesi* is of high quality. There is general consensus that most lepidopteran species have a sex chromosome system of WZ/ZZ (female/male), but the genus *Papilio* is a special group among lepidopterans, possessing both the WZ/ZZ (*P. machaon*) and Z/ZZ (*P. elwesi*) sex chromosome systems.

The mitochondrial genome of *P. elwesi* was assembled into a circularity with a size of 15,337 bp, and the length of the assembled mitochondrion was close to that of most *Papilio* species (~15,300 bp) public in the NCBI. The mitochondrion was annotated with 37 genes; the numbers of transfer RNA genes, protein-coding genes, and ribosomal RNA genes were 22, 13, and 2, respectively. A majority of the genes were found on the positive strand (majority strand or J-strand), including fourteen tRNA genes (tRNA-Met, tRNA-Ile, tRNA-Trp, tRNA-Leu, tRNA-Lys, tRNA-Asp, tRNA-Gly, tRNA-Ala, tRNA-Arg, tRNA-Asn, tRNA-Ser, tRNA-Glu, tRNA-Thr, and tRNA-Ser2) and nine protein-coding genes (ND2, COX1, COX2, ATP8, ATP6, COX3, ND3, ND6, and CYTB), with the remaining genes in the reverse strand (minority strand or N-strand). It displays the same gene order as the publicly available swallowtail butterflies deposited in the NCBI, such as *Papilio maraho* (NC_014055.1), *Papilio helenus* (KP247522.1), and *Papilio dialis* (OP135983.1). Moreover, its gene order and orientation are consistent with some other lepidopteran species, e.g., moths [76,77]. Nevertheless, the order of trnM–trnI–trnQ in *P. elwesi* is different from the ancestor of Ditrysia, which is trnI–trn–trnM [78], while the former undergoes rearrangement as trnM–trnI–trnQ. The base compositions of A, T, G, and C were 40.17%, 39.61%, 7.49%, and 12.73%, respectively, indicating a high A+T content (79.78%) in the mitochondrion of *P. elwesi*. The mitochondrial genome assembled and annotated in the study was deposited in GenBank under the accession number OQ581151

### 3.3. Genome Annotation

Among the 131.99 Mb (36.82%) repetitive elements masked in the *P. elwesi* genome, there were 57.37 Mb (16.00%) of long interspersed nuclear elements (LINEs), 4.08 Mb (1.14%) of short interspersed nuclear elements (SINEs), 12.00 Mb (3.35%) of long terminal repeat (LTR) elements, 9.16 Mb (2.56%) of DNA transposons, 8.70 Mb (2.43%) of simple repeats, 18.09 Mb (5.04%) of rolling circles, and 19.98 Mb (5.57%) of unclassified elements (Appendix A; Figure 1a). The proportion of repetitive elements was subequal to that of *P. machaon* (37.24%, data from NCBI *Papilio machaon* Annotation Release 100).

There were 1296 identified ncRNAs, which contained infrastructural (housekeeping) and regulatory ncRNAs. The numbers of ribosomal RNA (rRNA), transfer RNA (tRNA), small nuclear RNA (snRNA), microRNA (miRNA), long ncRNA (lncRNA), ribozyme, and other ncRNA were 357, 669, 76, 92, 2, 5, and 95, respectively (Appendix A). The rRNAs included 44 5.8S rRNAs, 162 5S rRNAs, 83 large subunit rRNAs (LSU), and 68 small subunit rRNAs (SSU). The tRNAs had 21 isotypes. The snRNAs were classified into seven snoRNAs (small nucleolar RNA; 6 CD-box, 1 HACA-box), and nine spliceosomal RNAs containing three minor ones. The miRNAs were classified into 55 families, ribozymes into 2 families, and the other ncRNAs into 5 families. The number of ncRNAs in the *P. machaon* genome was 1334, slightly more than that of *P. elwesi*.

The MAKER pipeline predicted 13,681 PCGs with a mean gene/protein length of 8529.8/571.7. The numbers of exons, introns, and CDSs per gene were 7.4, 6.4, and 7.2, respectively, and their corresponding mean lengths were 332.3, 1003.9, and 221.3 (Appendix A). The BUSCO achieved a completeness assessment of 98.60% (insecta_odb10, *n* = 1367) for the protein sequences, demonstrating predictions of high quality. A total of 13,442 (98.25%) genes were hit in the UniProtKB database with at least one record, and 11,414 (83.43%) and 13,042 (95.33%) were predicted by InterProScan and eggNOG, respectively. Genes with 10,208 GO items and 4995 KEGG pathway terms were identified by combining the InterProScan and eggNOG results (Appendix A).

### 3.4. Gene Family Evolution and Phylogenetic Relationships

OrthoFinder identified 156,231 genes among 11 species, including 150,006 (96.02%) genes assigned to 14,488 orthogroups and 6225 (3.98%) unassigned genes. There were 7650 orthogroups shared by all species, including 3891 single-copy orthogroups, and 5891 species-specific genes clustered into 1524 orthogroups (Appendix A; Figure 1c). The distribution of the gene families (orthogroups) for the 11 species is summarized in Appendix A. In the *P. elwesi* genome, 13,411 (98.03%) of the 13,681 identified genes were contained in 11,499 orthogroups, and the numbers of species-specific orthogroups and genes were 38 and 113, respectively (Appendix A).

The phylogenetic inference using 1508 single-copy orthologous genes (including 1,192,952 amino acids) recovered the monophyly of the Papilionoidea and *Papilio*. The reconstructed phylogeny of Papilionoidea was (Papilionidae + (Pieridae + (Nymphalidae + Lycaenidae))), and all nodes had absolute support (UFB/SH-aLRT = 100/100). The age of the most recent common ancestor (MRCA) of the Papilionoidea was 115.84 (106.44–125.25) Ma. The Pieridae originated from 101.15 (92.29–108.93) Ma, the Nymphalidae and Lycaenidae diverged from 89.19 (81.62–96.8) Ma, and the *Papilio* originated from 30.74 (28.15–33.53) Ma (Figure 1c). The results of the two analyses agree with those of Espeland et al. [7], Lu et al. [16], and Allio et al. [79].

The gene family evolution analysis in CAFE indicated that the global error estimation of the input data was 0.0875, and the estimated lambda was 0.00193608081496. In total, 621 gene families expanded and 863 gene families contracted in *P. elwesi*. Among them, 104 (77 expansions and 27 contractions) gene families experienced rapid evolution (Appendix A; Figure 1c). These rapid expansion families included those responsible for cuticle formation (cuticular protein: 17; chitin-binding type-2 domain-containing protein: 15; insect cuticle protein: 7), detoxification metabolism (glutathione S-transferase: 14; cytochrome P450: 5), digestion (trypsin: 11), juvenile hormone synthesis (farnesyl pyrophosphate synthase: 9), chemosensation (gustatory receptor: 8; PBP/GOBP family: 8; odorant receptor: 5), immunity (cecropin family: 7), and so on (Appendix A, Figure 1b). We speculate that these expanded families are related to foraging and adaptation to the host plant, as terpenoid-derived metabolites in Lauraceae trees, such as linalool, have recently been reported to act on the expressions of cuticular proteins and cytochrome P450s simultaneously [80]. The GO and KEGG enrichments of the rapid expansion families further indicate that their functions are involved in detoxification, metabolism, alarm pheromone, and some functions regarding biological processes for which we could not discover the specific meaning (Appendix A).

### 3.5. Chromosomal Synteny

Chromosome comparisons between the genomes of *P. elwesi* and *P. machaon* revealed conserved syntenic relationships among the chromosomes (Figure 1d). Eighty syntenic blocks with 20,067 (74.2%) collinear genes from the two genomes were identified.

Most chromosomes (1, 4, 6–10, 13, 14, 16–20, 22, 25, 29) of *P. elwesi* matched perfectly with the corresponding chromosomes of *P. machaon*, whereas no collinearity was found between chromosome 30 of *P. elwesi* (Pelw30) and chromosome 28 of *P. machaon* (Pmac28) (Figure 1d). We can confirm that Pelw30, with the interaction signals, is a complete chromosome belonging to *P. elwesi* according to the Hi-C heatmap (Appendix A). We suspect that Pelw30 is a neo-chromosome or W sex chromosome. Specifically, the high content of repetitive sequences in Pelw30 is particularly similar to that of the W sex chromosome [25,27,81], but there is no clear evidence for this at present. Chromosome 15 of *P. machaon* (Pmac15) was collinear to chromosomes 15 (Pelw15) and 28 (Pelw28) of *P. elwesi*. The Z sex chromosomes of the two papilionid species also showed collinearity, and the W sex chromosome of *P. machaon* (PmacW) was related to the Z chromosome of *P. elwesi* (PelwZ). The dominant role that the W chromosome plays in determining the sex of Lepidoptera seems to be doubtful, as demonstrated here, and has been raised by several previous studies (e.g., Yoshido et al. [82]; Dalíková et al. [26]; Fraïsse et al. [27]). This also implies that Z/ZZ may be an ancestral system, with the W chromosome evolving later from a common ancestor [26]. Taking this into consideration, the genome assembled here in high quality may serve as a genomic resource for future exploration and understanding of the sex determination mechanism and sex chromosome evolution in Lepidoptera.

## 4. Conclusions

With PacBio HIFi, Hi-C, ONT, and Illumina sequencing data, we report the lepidopteran genome assembly of *Papilio elwesi*, an endemic butterfly from the Chinese mainland. The chromosome-level assembly comprises 207 contigs with a size of 358.5 Mb. The BUSCO completeness of the assembly and the 13,681 predicted protein-coding genes reach 99% and 98.6%, respectively. The results of the phylogenomic analysis support Papilionidae at the basal position to all other butterflies. The sex chromosome system of *P. elwesi* is Z/ZZ (female/male), which differs from the majority of lepidopterans (WZ/ZZ). The high-quality genome assembled here is important evidence for revealing lepidopteran sex determination systems and an important source of data for future studies exploring lepidopteran evolution and comparative genomics.

## Figures and Tables

**Figure 1 insects-14-00304-f001:**
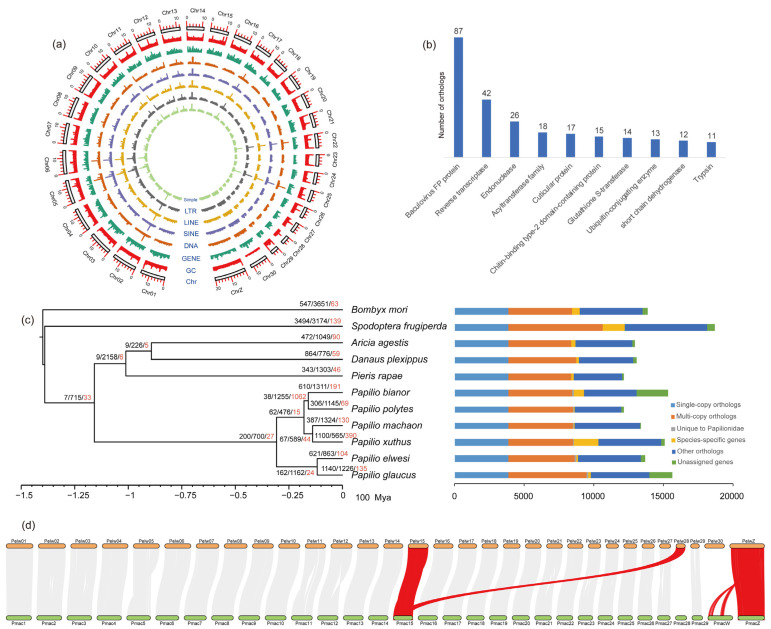
Genome characteristics, phylogeny, gene family evolution of *Papilio elwesi*, and interspecific synteny. (**a**) Genomic characters of each chromosome with its length (Chr), GC content (GC), protein-coding genes density (Gene), repetitive elements of DNA transposons (DNA), long interspersed nuclear elements (LINE), short interspersed nuclear elements (SINE), long terminal repeat elements (LTR), and simple repeats (Simple). (**b**) Phylogeny and gene family changes of 11 lepidopteran species. The values show the number of expanded, contracted, and significantly rapid evolutional (red) gene families of nodes and species. (**c**) The top 10 gene families with significant expansions. (**d**) Chromosomal synteny between *Papilio elwesi* (Pelw) and *Papilio machaon* (Pmac).

## Data Availability

All of the raw sequencing data (accessions CRR588447–CRR588454) and the final genome assembly (accession GWHBOVJ00000000) of *Papilio elwesi* have been deposited at the BioProject PRJCA012581 at the China National Center for Bioinformation-National Genomics Data Center (CNCB-NGDC). We have also uploaded all raw sequencing data (SRR23648870, SRR23689296, SRR23700075, SRR23702046, SRR23703133) and final genome assembly (JARFMI000000000) to the NCBI under BioProject PRJNA937921. Additionally, information regarding the genome annotation has been uploaded to Figshare and is available at https://doi.org/10.6084/m9.figshare.22045085.v1.

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
