# Peer review of "Chromosome-Level Genome Assembly of Papilio elwesi Leech, 1889 (Lepidoptera: Papilionidae)"

_insects, 2023, doi:10.3390/insects14030304_

Round 1

Reviewer 1 Report

The paper describe the genome assembly at Chromosomal level in Papilio elwesi. Overall the paper is well done and clear and there are no major issues about the methods (I gave my comments below). The only issue is that based on the current introduction it is not clear why this paper is important for science. So I suggest the author to expand the introduction explaining why they choose this species and why it is so important. This will increase the visibility of the paper. Another issue is that the authors did not described the mitochondrial genome, did they removed from the assembly? They should clarify this aspect.

Introduction:

The aims of the paper are not very well described, the author could give some additional information why they sequenced the genome of this exact species, why it is so important compare to other species. 

Material and methods:

I can agree on the use of a pupae for the whole genome sequencing, but for transcriptome I found a quite wrong decision, because pupae is a key stage in development during which only few genes are expressed that are mostly important for organisms development. I would be not surprise if many important genes can not be annotated.

Is there any quality control (like agilent or Femto) for the whole genome and RNA extraction before sequencing?

The authors did not described the mitochondrial genome. Based on the sequencing methods they should have identified and either removed and/or described.

Section 2.4: Ortofinder is working only with proteins. The authors used the proteomes of 10 species, did they also extracted the sequences from the Proteome of the study species (like using search by homology)? Or only used the predicted sequences from Augustus? For phylogeny purposes it would be better using OrthoDB.

Section 3.2. The authors described the number of chromosomes and genome size based on reads. Do the author have any information about the estimated genome size based on FACS and on the correct number of chromosomes?

Could be that there is one less chromosome than expected based on other species (Figure 1d)?

Phylogenetic tree, I would like to see nodes supports and to know how much of the initial aligned genes have been trimmed. 

Figure 1c I can not follow the number on the nodes, this should be described in the figure legend.

Author Response

Introduction: The aims of the paper are not very well described, the author could give some additional information why they sequenced the genome of this exact species, why it is so important compare to other species. Reply: We added the relevant content. Material and methods: I can agree on the use of a pupae for the whole genome sequencing, but for transcriptome I found a quite wrong decision, because pupae is a key stage in development during which only few genes are expressed that are mostly important for organisms development. I would be not surprise if many important genes can not be annotated. Reply: It is difficult to have the opportunity to encounter and collect specimens of Papilio elwesi, and we only have pupae. Considering that only the transcriptome of the pupa is available, we integrated other evidence in predicting genes, including Arthropoda protein sequences from OrthoDB10 and high-quality assemblies and annotations of six reference species. Is there any quality control (like agilent or Femto) for the whole genome and RNA extraction before sequencing? Reply: Yes, we performed quality control for the extracted DNA and RNA before sequencing. For DNA, we used NanoDrop 2000 Spectrophotometer, Qubit fluorometer, Pulsed field electrophoresis (Bio-rad CHEF) and agarose gel electrophoresis. For RNA, we used Agilent 4200 TapeStation system and NanoDrop 2000 Spectrophotometer. And relevant content was also added to the text. The authors did not described the mitochondrial genome. Based on the sequencing methods they should have identified and either removed and/or described. Reply: We had assembled its mitochondrial genome. And we have added it to the text. Section 2.4: Ortofinder is working only with proteins. The authors used the proteomes of 10 species, did they also extracted the sequences from the Proteome of the study species (like using search by homology)? Or only used the predicted sequences from Augustus? For phylogeny purposes it would be better using OrthoDB. Reply: Yes, our study species, Papilio elwesi, was included in the 11 selected species for orthology inference with diamond searching in OrthoFinder v2.5.2. Overall, OrthoFinder is better than OrthoDB, based on a comparison between them, please see Fig. 3 in “OrthoFinder: solving fundamental biases in whole genome comparisons dramatically improves orthogroup inference accuracy” (Emms & Skelly2015). The updated OrthoFinder2 has been further improved. Section 3.2. The authors described the number of chromosomes and genome size based on reads. Do the author have any information about the estimated genome size based on FACS and on the correct number of chromosomes? Reply: We did not use FACS because the genome size we assembled was approximate to the estimated genome size, also approximate to other published butterfly genomes, and had high genome completeness and low heterozygosity.

Reviewer 2 Report

It is good progress to assemble a genome of a lepidopteran species having ZO sex-determination system. The remaining problem is that no collinearity was found for Pelw30 and Pmac28. Either or both of the genome assemblies might be incorrect for the chromosomes. I strongly recommend the authors to map single-copy conserved genes of Pelw30 and Pmac28 onto the B. mori genome. In addition, GC-content of Pelw30 seems to be consistently low compared with other chromosomes (Fig. 1A). A possible explanation is needed for that. 

Author Response

Reply: We can confirm that the chromosome Pelw30 is not contaminated. Although we thought that the depth of sequencing of chromosome Pelw30 was abnormal, we could find that this chromosome interacted with the whole genome through the Hi-C heatmap (see fig. S2). Pelw30 has strong interaction signals inside and no pollution signals, so it is a complete chromosome; it also interacts with other chromosomes, just not as strongly, so Pelw30 belongs to the species Papilio elwesi. Moreover, we blasted chromosome Pelw30 at GenBank and did not find any contamination. We did not find collinearity between Pelw30 and B. mori. The low GC content of Pelw30 is due to the high content of repetitive sequences and transposons (please enlarge fig. 1a). As a result, its sequencing depth cannot be given accurately. We suspect that this Pelw30 is a neo-chromosome or sex chromosome W. The high proportion of repetitive sequences of Pelw30 is especially like the performance of the sex chromosome W, but its sequencing depth is not. Regarding the formation of sex chromosome W, there are several theories in Lepidoptera; W comes from various sources, while Z has always been conservative (Sahara, Yoshido and Traut 2012; Dalíková et al., 2017; Fraïsse, Picard and Vicoso 2017; Dai et al., 2022). Current evidence cannot determine that it is chromosome W or neo-chromosome, but we can be sure that it is a complete chromosome of P. elwesi. And it needs to be revealed by further comparative genomics analysis in the future.

Round 2

Reviewer 1 Report

I found the paper strongly improved. I am still not convinced that using the pupae for RNAseq was appropriate, but the authors can still improve this aspect in the future.

Anyway, I believe the paper is now ready for publication.